# Safety Assessment of an Oral Therapeutic Dose of Firocoxib on Healthy Horses

**DOI:** 10.3390/vetsci10090531

**Published:** 2023-08-22

**Authors:** Renatha A. Araújo, Nathali A. A. Sales, Roberta C. Basile, Walter H. Feringer-Junior, Maricy Apparício, Guilherme C. Ferraz, Antonio Queiroz-Neto

**Affiliations:** 1Laboratory of Equine Exercise Physiology and Pharmacology (LAFEQ), Department of Animal Morphology and Physiology, School of Agricultural and Veterinary Studies, São Paulo State University, FCAV/UNESP, Via de Acesso Prof. Paulo D. Castellane s/n., Jaboticabal 14884-900, SP, Brazil; agassi.sales@unesp.br (N.A.A.S.); rcbasile@gmail.com (R.C.B.); guilherme.c.ferraz@unesp.br (G.C.F.); a.queiroz-neto@unesp.br (A.Q.-N.); 2Department of Veterinary Medicine, Metropolitan University of Santos, UNIMES, Av. Gen. Francisco Glicério, 8, Santos 11045-002, SP, Brazil; walterferinger@gmail.com; 3Department of Veterinary Surgery and Animal Reproduction, School of Veterinary Medicine and Animal Science, São Paulo State University (UNESP), Rua Professor Doutor Walter Mauricio Correa, s/n., Botucatu 18618-681, SP, Brazil; maricy.apparicio@unesp.br

**Keywords:** firocoxib, coagulation, gastroscopy, hematology, selective COX-2 inhibitor, drug safety

## Abstract

**Simple Summary:**

This study was conducted to evaluate the effects of oral firocoxib on healthy horses. The drug is a non-steroidal anti-inflammatory medication routinely used in veterinary medicine. The study found that administering firocoxib to the horses did not cause any adverse effects on the stomach mucosa. However, it slightly reduces the blood’s ability to clot, which was unexpected since the drug was supposed to promote clotting. Overall, firocoxib was deemed safe for horses and can be used to manage inflammation in these animals. Clinicians should be conscious of potential anti-coagulopathy secondary to firocoxib administration with long-term use or critical illness.

**Abstract:**

Firocoxib is a non-steroidal anti-inflammatory drug specifically formulated for veterinary medicine and selectively acts on inhibiting the cyclooxygenase 2 enzyme (COX-2). This study evaluated the possible adverse effects of administering oral therapeutic firocoxib on gastric mucosa, hematological parameters, coagulation cascade, and hepatic and renal biochemistry in healthy horses. Nine clinically healthy Arabian horses, approximately 9 years old, received 0.1 mg/kg of oral firocoxib for 14 days. The gastroscopic examination was conducted 1 day before starting treatment (D0) and two days after the last blood collection (D23). Venous blood samples were obtained for laboratory tests on day 1, immediately prior to the initiation of treatment (D1), after 7 and 14 days of treatment (D7 and D14), and 7 days after the conclusion of treatment (D21. No changes were found in the gastroscopic and hematological tests. Coagulation and serum biochemistry levels remain between these species’ average values. However, the increased activated partial thromboplastin time (aPTT) and prothrombin time (PT) indicate reduced blood coagulation capacity, which contradicts the expected effect of treatment with selective COX-2 inhibitors, as these drugs theoretically promote coagulation. Administering firocoxib to horses is safe as it does not cause significant adverse reactions. Therefore, it is a suitable option for managing inflammatory conditions in these animals with attention to an unexpected adverse anti-coagulopathy effect, and further study is warranted.

## 1. Introduction

Among the diseases that affect the equines, musculoskeletal disorders occupy the leading position. They are the diseases with the highest prevalence in the equine species, besides being the most frequent cause of the termination of the sports career of these animals [1]. The training routine may lead to the development of injuries that often require treatment with anti-inflammatory drugs. Certain diseases, such as osteoarthritis and other chronic or difficult-to-control inflammatory conditions, necessitate extended treatments, potentially resulting in adverse effects due to the non-specific inhibition of cyclooxygenases.

Steroidal and non-steroidal anti-inflammatory drugs (NSAIDs) can manage inflammatory processes, their mediators, and pain. Nonsteroidal anti-inflammatory drugs (NSAIDs) are widely used drugs worldwide, both in human and veterinary medicine. They exhibit diverse chemical structures; however, they share similar mechanisms of action, acting through the inhibition of phosphodiesterase A2 and cyclooxygenases (COX), resulting in anti-inflammatory effects. This action prevents the production of prostaglandins (PG) and thromboxanes (TX) [2].

Based on their ability to inhibit COX-1 or COX-2, NSAIDs can be classified into three groups: non-selective inhibitors, such as aspirin, phenylbutazone, ibuprofen, ketoprofen, and dipyrone; preferential COX-2 inhibitors, such as meloxicam, carprofen, nimesulide, celecoxib, and selective COX-2 inhibitors, such as valdecoxib, rofecoxib, lumiracoxib, and firocoxib [3]. Most NSAIDs used in veterinary therapeutics act non-specifically on both COX-1 and COX-2, and their use is associated with the emergence of adverse effects, as the inhibition of COX-1 interferes with various physiological processes. Selective COX-2 inhibitors exhibit high anti-inflammatory potential and reduced toxic effects on organic systems. However, the chronic use of selective inhibitors has been linked to the development of cardiovascular disorders, blood coagulation disturbances [4], and alterations in renal perfusion and glomerular filtration [5]. Even before the studies that accurately defined the distinct isoforms of cyclooxygenase and their functions, the difference in gastrointestinal tract safety among NSAIDs was already observed. With the advancement of further research, it became evident that the NSAIDs causing the least impact on the gastrointestinal tract predominantly inhibit COX-2 [6].

COXs can be found in all mammalian tissues [7]. Both isoforms exhibit highly similar molecular structure and molecular weight, with approximately 60% amino acid homology [8]. The expression of COX-1 and COX-2 varies among different cells in the organism, as well as the functions of the PGs derived from the breakdown of arachidonic acid (AA) by each of them.

In humans, the chronic use of selective COX-2 inhibitors is associated with an increased risk of cardiovascular disorders such as arterial hypertension and cerebral thrombosis [4,9,10]. These inhibitors prevent the production of prostacyclin (PGI2), specifically in the cells lining the blood vessels, the vascular endothelium, without reducing the production of thromboxane A2 (TXA2) by platelets, leading to an imbalance in vasodilation/vasoconstriction mechanisms and coagulation. This effect is not as pronounced with preferential COX-2 inhibitors, as these drugs also maintain some level of COX-1 blockade, although at lower levels than COX-2 inhibition, allowing for maintaining the PGI2/TXA2 balance. The analysis of coagulation factors can be a helpful tool to evaluate the emergence of adverse effects from the administration of selective COX-2 inhibitors and other non-steroidal anti-inflammatory drugs.

Firocoxib is a non-steroidal anti-inflammatory drug (NSAID) indicated specifically for veterinary use, designed to target COX-2 enzymes associated with inflammation. In contrast to other medications that also interfere with COX-1, firocoxib exerts minimal impact on this enzyme, thereby reducing the risk of adverse side effects [11]. This drug acts as a potent and highly selective inhibitor of the COX-2 isoform. In an in vitro study using whole blood from dogs, firocoxib demonstrated selectivity 350 to 430 times greater for COX-2 than for COX-1 [12]. Moreover, firocoxib exhibits a long half-life, allowing for once-daily administration, making it favored for treating horse pain [13]. It finds common usage in equine clinics to address various types of inflammation, whether systemic or localized [11,14].

The dosage of firocoxib used will depend on the species being treated. According to FDA recommendations the dosage for dogs is 5 mg/kg every 24 h, and for horses, it is 0.1 mg/kg [15,16]. Firocoxib is metabolized in the liver and excreted by the kidneys, and it is administered orally in horses. Following oral administration, the drug reaches absolute plasma bioavailability after 3.9 h, where it strongly binds to plasma proteins. The long half-life of firocoxib (30 h) allows for once-daily administration, which is sufficient to maintain adequate plasma levels for pain control [7].

Among the most common adverse effects observed with administering non-selective NSAIDs in horses, delayed healing of the enteric mucosa is a prominent concern compared to selective COX-2 inhibitors [17,18,19]. Inhibiting COX-2 can help prevent pain and inflammation while promoting mucosal tissue repair by allowing COX-1 to function correctly [20]. Studies suggest that the use of NSAIDs leads to delayed healing of gastric ulcers [21] since both angiogenesis and cell proliferation are involved in this process [22].

Gastroscopy is considered the safest method to evaluate the presence of gastric lesions [23]. Hematological evaluation allows for detecting changes in red blood cell count, hemoglobin concentration, and hematocrit, usually seen as decreased count levels resulting from any gastrointestinal lesions that may occur during therapy with such medication [24]. Coagulation tests such as Prothrombin Time (PT) and Activated Partial Thromboplastin Time (APTT) and the measurement of fibrinogen can help detect alterations in hemostasis mechanisms [25]. Serum biochemical evaluation allows access to hepatic and renal metabolism alterations that may result from therapy with anti-inflammatory drugs.

Although selective COX-2 inhibitors appear to cause fewer adverse effects than traditional anti-inflammatory drugs, safety studies are necessary to establish the best treatment regimens with these new drugs. Similar to COX-1, COX-2 is also expressed constitutively in tissues, and its pharmacological suppression can lead to alterations.

Regarding firocoxib, there is a lack of literature data supporting hypotheses on minimizing toxicity resulting from its use, especially in the equine species. In this context, it is possible to utilize clinical and laboratory tools to observe the responses to treatment with NSAIDs. As a treatment option for non-infectious joint diseases in horses, the therapy with firocoxib aims to mitigate the adverse reactions commonly observed with non-selective inhibitors. This study evaluated the safety of administering firocoxib orally to healthy horses for 14 days.

## 2. Materials and Methods

### 2.1. Ethics in the Use of Animals

All procedures followed the Ethical Principles in Animal Experimentation adopted by the National Council for Control in Animal Experimentation (CONCEA). The protocol was reviewed and approved by the institutional Ethics Committee for the Use of Animals (CEUA, UNESP, Jaboticabal, Brazil) and registered under Protocol no. 015002/10.

### 2.2. Animals

Nine horses, four of which were castrated males, and five non-pregnant females, with an average body mass of 405 ± 31 kg and an average age of 11 ± 3 years, belonging to the experimental herd of the Laboratory of Equine Pharmacology and Physiology (LAFEQ) at the Faculty of Agricultural and Veterinary Sciences—UNESP, Jaboticabal Campus. The animals were kept in paddocks during the testing period and received a balanced diet, Tifton hay, and ad libitum water. They were considered healthy based on clinical and laboratory evaluations; all animals underwent deworming, vaccination, and ectoparasite control programs before the beginning of the experimental period.

### 2.3. Procedures

The horses received once-daily oral administration of 0.1 mg/kg of firocoxib for 14 days. The tablets were crushed using a mortar and pestle and weighed on a precision balance to obtain the exact amount of drug for each animal. The administration was performed by adding 10 mL of corn syrup to the drug, and the resulting paste was provided to the animals orally using a 20 mL syringe with the tip cut off. Gastric endoscopy was performed on all animals one day before the start of the treatment (D0) and two days after the last data collection (D23). Blood samples were collected for laboratory tests on the following days: D1, before the start of the treatment; D7, 7 days after the start of the treatment; D14, 14 days after the beginning of the treatment, and D21, 1 week after the end of the therapy. Daily observations were made regarding the general health status of the animals, including the appearance of discomfort, decreased appetite, and diarrhea.

### 2.4. Gastroscopy

The animals were fasted for four hours before each evaluation as part of the experimental procedure, followed by gastric emptying through nasogastric intubation, water infusion, and siphoning of the content. After this procedure, the animals were sedated with detomidine hydrochloride at a dose of 0.015 mg/kg administered intravenously. The animals were restrained in a stock. The gastroscopies were conducted using a video gastroscope with a length of 300 cm and a diameter of 0.8 cm, coupled to a xenon light source (Endoscope Karl Storz GMBH & CO, Tuttlingen, BW, Germany). The probe’s tip was lubricated with lidocaine gel, introduced through the nostril, and directed to reach the stomach. Air was insufflated to distend the gastric cavity to visualize the structures better. The non-glandular mucosa, margo plicatus, and visible portion of the glandular mucosa were observed to detect possible ulcers and other clinically relevant gastric alterations. The classification of observed lesions was performed according to McAllister [23]: 0 = no lesion, 1 = inflammation only (intact mucosa), 2 = superficial lesions (mucosal rupture only), 3 = moderate lesion (involvement of deeper tissue), 4 = deeper tissue involvement with evident bleeding, and 5 = close to perforation. After the evaluation, the endoscope was gently withdrawn while the insufflated air in the stomach was released.

### 2.5. Hematological Evaluation

Blood samples of 5 mL were collected from a vein using tubes with negative pressure containing ethylenediaminetetraacetic acid (EDTA). The global counts of red and white blood cells, hemoglobin concentration, hematocrit, and platelet count were performed using a veterinary hematology analyzer (Vet Auto Hematology Analyzer^®^ BC-2800 VETMindray, Shenzen, China). The differential white blood cell count was conducted through a blood smear.

### 2.6. Coagulation Evaluation

After collecting 5 mL of blood in tubes with 3.8% sodium citrate, it was promptly centrifuged. The resulting plasma was utilized for coagulation tests, which included measuring prothrombin time, activated partial thromboplastin time, and fibrinogen concentration. These tests were conducted using the methodology specified for the commercial reagents (Kits Labtest^®^ Lagoa Santa, MG, Brazil) used in a coagulometer (Quick Timer^®^ II Drake, São José do Rio Preto, SP, Brazil), activated partial thromboplastin time, and fibrinogen concentration according to the methodology determined for commercial reagents used in a coagulometer.

### 2.7. Biochemical Evaluation

With the help of negative-pressure tubes containing a clot activator, 10 mLs of blood was collected and kept at room temperature for 15 min, then centrifuged for 10 min at 1300× *g*. The separated serum was pipetted and used for serum biochemical evaluation. Serum activities of ALP, ALT, AST, LDH, GGT, CK, and serum concentrations of urea, creatinine, and TP were evaluated. We conducted the analyses by utilizing a spectrophotometer (Labquest^®^ Bio 2000, Barueri, SP, Brazil), that is, semi-automatic and commercial colorimetric reagents (Kits Labtest^®^ Lagoa Santa, MG, Brazil). All blood samples were obtained through jugular venipuncture.

### 2.8. Statistical Methods

Statistical analysis was conducted on Sigma Plot^®^ 11.0 program (Systat Software Inc., San Jose, CA, USA). First, the data were tested for normal distribution using the Shapiro-Wilk test and then compared using repeated measures analysis of variance (ANOVA). The rejected hypotheses were subjected to the Tukey test. All tests were conducted at a significance level of 5%.

## 3. Results

### 3.1. Gastroscopic Evaluation

During the initial gastroscopic evaluation (D0), all animals were classified with a score of 0 and showed no notable changes. After 14 days of treatment, none of the animals displayed any alterations, such as mucosal inflammation or ulcerative lesions (Figure 1). The most frequent location is within the non-glandular mucosa, along the margo plicatus on the lesser curvature. The margo plicatus serves as a highly exposed transitional area to gastric acid, where ulcers are often linked to the administration of NSAIDs [23]. The antrum is the terminal region of the glandular portion of the stomach, located closer to the pyloric sphincter, and is highly susceptible to developing gastric ulcers [26]. The pyloric region could not be visualized during the gastroscopic examination. The post-treatment investigation also resulted in a score of 0, indicating no difference between the evaluations. Therefore, there was no need for statistical analysis since the score of 0 was maintained.

### 3.2. Complete Blood Count (CBC)

In the hematological evaluation, only the leukocyte count and hemoglobin concentration had differences in the comparison of means between the evaluated time points throughout the experimental period (*p* < 0.05), as observed in Table 1. The leukocyte count decreased at D14 compared to D1 and returned to baseline values at D21, seven days after treatment. The hemoglobin concentration increased at D14 and D21 compared to D1. All hematological variables remained within the reference values for the species [27].

### 3.3. Coagulation Evaluation

In the coagulation study, all analyzed variables showed significant alterations (*p* < 0.05); as shown in Figure 2, the activated partial thromboplastin time (APTT) increased at D7, D14, and D21 when comparing the means to the mean of D1 Prothrombin time (PT) increased at D7, D14, and D21 compared to D1. Plasma fibrinogen concentration increased at D21 compared to D7 and D14. Platelet count decreased at D7, D14, and D21 compared to the D1 time point.

### 3.4. Biochemistry Values

In the evaluation of serum biochemistry, all researched variables remained within the reference values for equine species (AST—226 to 366 U/L; GGT—4.3 to 13.4 U/L; ALT—3 to 23 U/L; CK—2.4 to 23.4 U/L; ALP—143 to 395 U/L; LDH—162 to 412 U/L; Ure—21 to 51 mg/dL; Creatinine—1.2 to 1.9 mg/dL; Total protein—5.2 to 7.9 g/dL). [27]. There was an increase in AST activity at D14 compared to D1 (Figure 3A). GGT activity decreased at D21 compared to D14. However, there was no decrease compared to D1 and D7 (Figure 3B). ALP activity increased at D7 and D14 compared to D1 and returned to baseline values at D21. (Figure 3E). Serum activities of ALT, LDH, and CK did not undergo significant changes during the experiment.

The urea concentration decreased at D14 compared to D1 (Figure 4A). Creatinine remained stable during the treatment period (Figure 4B). However, at D21, it decreased compared to D1, D7, and D14. Total protein values did not show significant differences (Figure 4C).

## 4. Discussion

Gastrointestinal toxicity is critical limiting factor in NSAID therapy. Non-selective traditional NSAIDs impact the gastrointestinal tract by blocking COX-1 and, consequently, hindering the formation of gastroprotective PGs responsible for regulating HCl synthesis, adequate gastric mucosa perfusion, and mucous and bicarbonate production [29]. Caution is necessary when using selective COX-2 inhibitors in animals with pre-existing gastric ulcers because COX-2 seems to participate in an adaptive cytoprotective mechanism established in the presence of gastric ulcers and accelerates the healing process [21].

The gastrointestinal evaluation did not reveal any significant alterations. This finding reinforces that selective COX-2 inhibitors cause less damage to the gastrointestinal tract (GIT) than non-selective NSAIDs [20]. In a study with mules comparing the impact of three non-selective NSAIDs on the GIT, the observed alterations in treated animals included hyperemia, erosion, and ulceration in the gastric mucosa [30]. However, firocoxib did not produce gastric lesions in dogs treated for 90 days [23]. In a study conducted with horses, administering firocoxib at 0.1 mg/kg in an oral paste formulation caused oral ulcers. The healing process of pre-existing oral ulcers during treatment was carried out for 42 days [31].

When using non-selective NSAIDs in therapy, it is common to observe a decrease in red blood cell count, hemoglobin concentration, and hematocrit [30,32], and mainly due to how the GIT responds to COX-1 inhibition, a reduction in gastroprotective prostaglandin production. Additionally, the anticoagulant action of COX-1 inhibitors can facilitate gastrointestinal bleeding, especially in the presence of ulcers, as they prevent the formation of TXA2, which is involved in hemostasis [33]. However, these alterations were not observed in the present study, suggesting that firocoxib did not cause injuries capable of reducing the red blood cell count. This finding is consistent with another experiment comparing the efficiency of firocoxib with phenylbutazone in animals with lameness, where the mean hematological indices remained within the reference values for both drug-treated groups [16].

Leukocytosis can be developed in response to treatment with NSAIDs [24]. According to the online platform HealthMe, which provides information about side effects and interactions between medications based on real medical data collected from sources such as FDA reports and other health databases, 0.29% of patients treated with celecoxib exhibited leukopenia [34]. Celecoxib also causes leukopenia in dogs treated for 20 days [35]. The reduction in the overall leukocyte count in our study did not represent a significant alteration in the differential count, like the findings in a study conducted by the FDA in 2010 [16], where horses treated with firocoxib for 14 days had a decrease in total leukocyte and neutrophil counts. The group treated with phenylbutazone also showed increased basophils during treatment.

The inhibition of COX-2 combined with the maintenance of COX-1 activity can cause an imbalance in the proportions of PGI2/TXA2, promoting increased blood coagulation and favoring the formation of blood clots [36]. APTT measures the activity of the intrinsic coagulation pathway [37]. PT evaluates the activity of the extrinsic coagulation pathway. Together, the increase in APTT and PT may suggest hepatic problems in the formation of coagulation factors, as well as fibrinogen deficiency [38]. Studies conducted with dogs treated with meloxicam [39] and celecoxib [35] showed no increase in APTT and PT. In the present study, the rise in APTT and PT indicates a reduced capacity for blood coagulation, which goes against the expected response to treatment with selective COX-2 inhibitors since these drugs theoretically favor coagulation [36]. Plasma obtained using citrate tubes with an increased APTT and PT was reported due to storage time [40]. The reduction in platelet count may have caused a decrease in thromboplastin concentration, leading to an increase in APTT.

The use of a series of drugs, such as heparin, certain alkaloids, penicillin, sulfonamides, diuretics, anticonvulsants, and NSAIDs [41], can cause thrombocytopenia and it occurs due to the binding of drug-dependent antibodies to receptors present on the surface of platelets, leading to their destruction. However, this mechanism still needs to be clearly described, as it is an idiosyncratic reaction and only affects a small fraction of patients under therapy with sensitizing drugs [42]. Therefore, the decrease in platelet count observed in the present study does not corroborate with what is reported in the literature [6].

Fibrinogen is a soluble hepatic glycoprotein involved in the final stage of coagulation [37]. It also acts as an acute-phase protein, with its increase being associated with the onset of inflammatory processes [43] or the nutritional status of horses [44]. In our study, the increase in fibrinogen seems to have no clinical relevance as the values remained within the reference range for the equine species [45] and considering that the animals were untreated for seven days, it is improbable that the increase is related to the administration of firocoxib. In studies conducted using firocoxib [35,39], the treatment did not cause alterations in fibrinogen concentration. Another study conducted with horses comparing the administration of firocoxib and flunixin meglumine showed fibrinogen results within the reference values for the species throughout the experimental period. However, there was a significant increase in fibrinogen values when comparing groups during and after treatment [40].

An increase in fibrinogen was observed in a study that evaluated the effects of flunixin meglumine, meloxicam, and firocoxib in horses. The researchers noted that castration in the groups treated with the three drugs led to this increase [46]. In another study, it was observed that plasma concentration increased three days after the feeding procedure [47]. A different outcome emerged from a study that involved castration and ovariectomy; in this case, the increase in fibrinogen occurred immediately after the procedures [48]. Fibrinogen is classically defined as an acute-phase protein in most species. However, its increase may be associated with inflammation, suppurative situations, trauma, and cases of neoplasia [49].

The use of NSAIDs can cause hepatic injuries. While the exact cause is unknown, higher levels of the drug in the liver can create harmful byproducts and damage the mitochondria [50]. The increase in serum activity of AST and ALT indicates hepatocellular damage. A study conducted on dogs treated with firocoxib for 28 days found that there was an increase in ALT levels [51]. However, this increase was still within the normal range for the species. In the present study, an increase in AST was observed at the end of the treatment period. However, caution is needed when evaluating AST alone, as it is not a specific hepatic enzyme [52]. The primary route of elimination is facilitated by the liver through dealkylation and glucuronidation processes, resulting in the formation of inactive firocoxib metabolites [7]. This phenomenon might have taken place in the current study.

When the mitochondrial membrane becomes more permeable, it causes the plasma activities of AST and CK to increase [53]. These markers are better indicators of muscle injury than LDH. The assessment of muscle injury resulting from using NSAIDs is more relevant in the case of drugs whose primary route of administration is intramuscular, as in this case, an increase in CK can occur in response to muscle necrosis at the injection site [54]. In the present study, CK remained unchanged throughout the experimental period, suggesting that no injury was significant enough to increase its serum concentration. Cholestasis has been reported in human patients treated with celecoxib [55,56]. The increase in ALP and GGT values observed in the present study does not indicate cholestasis, as even the variables that showed significant changes remained within the reference values for the equine species [26,27], and the GGT values found in cholestatic conditions in horses are generally about 15 times higher than average [57].

Research has shown that selective COX-2 inhibitors do not offer any advantage in reducing the risk of renal toxicity compared to traditional NSAIDs, as both COX isoforms are present in healthy renal tissue, as confirmed by literature [5]. The reduction in plasma urea and creatinine levels has no clinical significance, as they do not reach the minimum limits for the equine species and are not accompanied by significant changes in liver enzymes. In dogs with osteoarthritis, treatment with firocoxib for 90 days did not change urea and creatinine concentrations or urine density [31]. Horses treated with firocoxib for 14 days did not show alterations in renal function [58]. However, when horses were treated with both phenylbutazone and firocoxib for 10 consecutive days, there was a notable rise in their serum creatinine levels, according to the literature [59].

Regarding urine, a study conducted with horses afflicted by conditions necessitating NSAIDs usage over a period of 3 to 5 days revealed the potential effects of flunixin meglumine, metamizole, and phenylbutazone on the proximal tubules [60]. Another study involving the administration of firocoxib to horses for 92 days at a dose of 0.25 mg/kg, utilizing a formulation similar to, albeit not distinct from, the final one, exhibited that one animal experienced elevated levels of mutant gamma-glutamyl transferase and protein after 28 days of treatment. In addition, during necropsy, this horse displayed lesions consistent with NSAID-induced toxicity and renal hemorrhage [16]. Furthermore, the prolonged use of phenylbutazone can significantly affect the ascending portion of the loop of Henle and cause increased inspiration of the urinary mucosa, resulting from the mechanism of protection against the necrotizing action of this drug [60].

During the firocoxib treatment, there were no changes in the serum concentration of total proteins, which particularly reinforces the absence of hepatotoxicity. This result is like the one found in a study with Wistar rats, where even with treatment doses five times higher than the suggested dosage for the species, no increase in total proteins was observed [32].

The lipophilic nature of firocoxib justifies its extensive distribution throughout organic tissues, including penetration into synovial fluid [7]. Thus, this drug is indicated for treating joint-related conditions in horses. A study involving horses naturally affected by osteoarthritis demonstrated the efficacy of firocoxib, showing that its anti-inflammatory potential is comparable to phenylbutazone [61]. The possibility of using the drug in horses for an extended period without observing toxic effects [62] supports its use in controlling chronic conditions, especially those of joint origin, which often require prolonged treatments.

## 5. Conclusions

Based on the study’s findings, when administering the recommended FDA dose of firocoxib for 14 days, no significant alterations were observed in the analyzed variables. These results suggest that the drug is well-tolerated and does not cause any notable physiological changes under these conditions in equine subjects.

## Figures and Tables

**Figure 1 vetsci-10-00531-f001:**
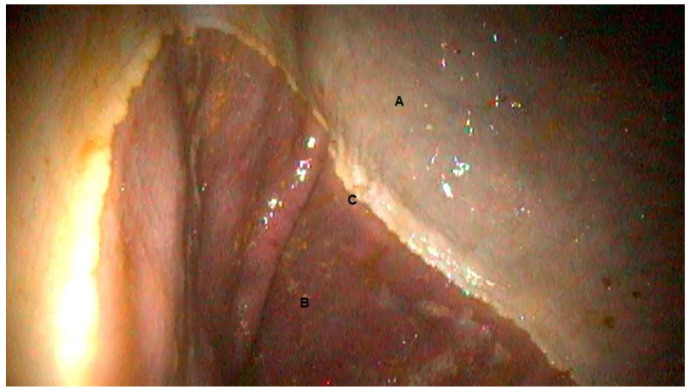
Gastroscopic image showing the non-glandular gastric mucosa (**A**), the glandular mucosa (**B**), and the plicatus margin (**C**).

**Figure 2 vetsci-10-00531-f002:**
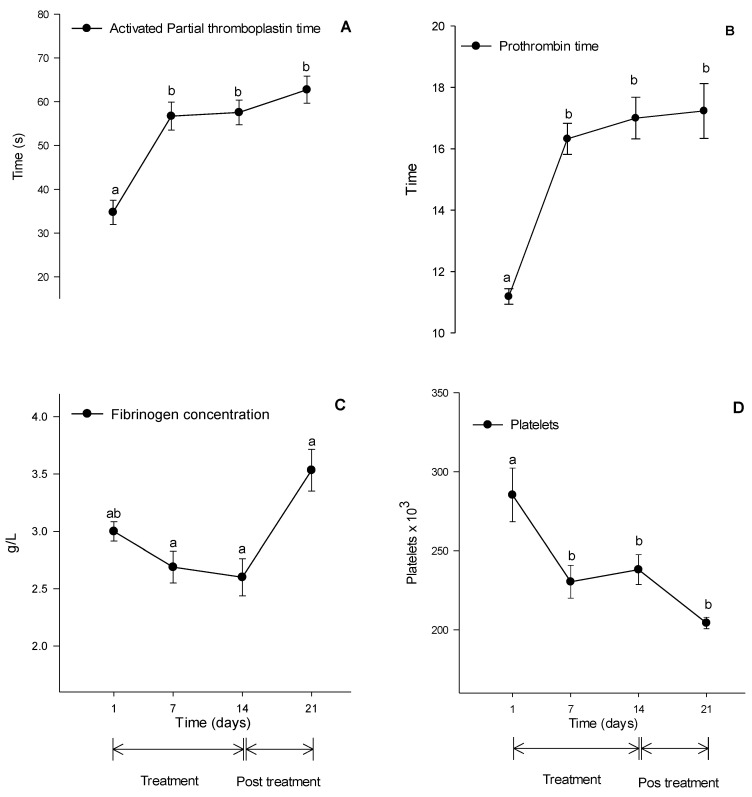
Means ± standard error of activated partial thromboplastin time (APTT) (**A**), prothrombin time (PT) (**B**), fibrinogen concentration (**C**), and platelet count (**D**). Different lower-case letters (a or b) indicate significant differences in mean comparisons by the Tukey test. Reference range for horses-APTT—30 to 44 s; PT —8.5 to 9.5 s; Fibrinogen—2.0 to 3.75 g/L; Platelets—75 to 300 × 10^3^/µL [28].

**Figure 3 vetsci-10-00531-f003:**
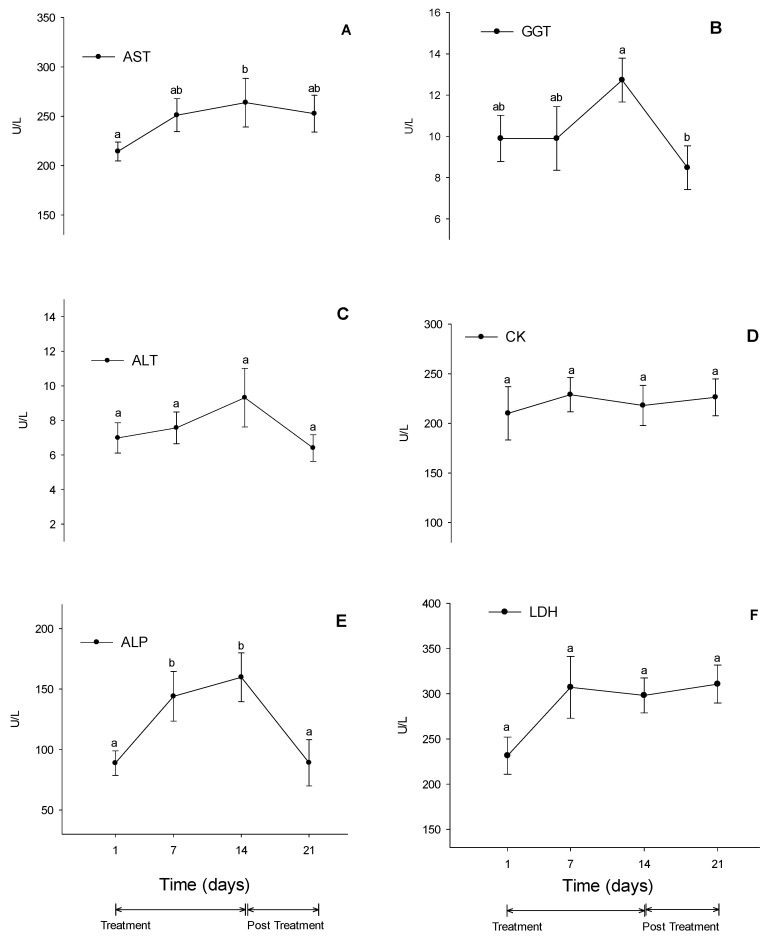
Serum concentrations of AST (**A**), GGT (**B**), ALT (**C**), CK (**D**), ALP (**E**), and LDH (**F**) enzymes. Different lower-case letters (a or b) indicate significant changes in mean comparisons by the Tukey test. Values are expressed as mean ± standard error.

**Figure 4 vetsci-10-00531-f004:**
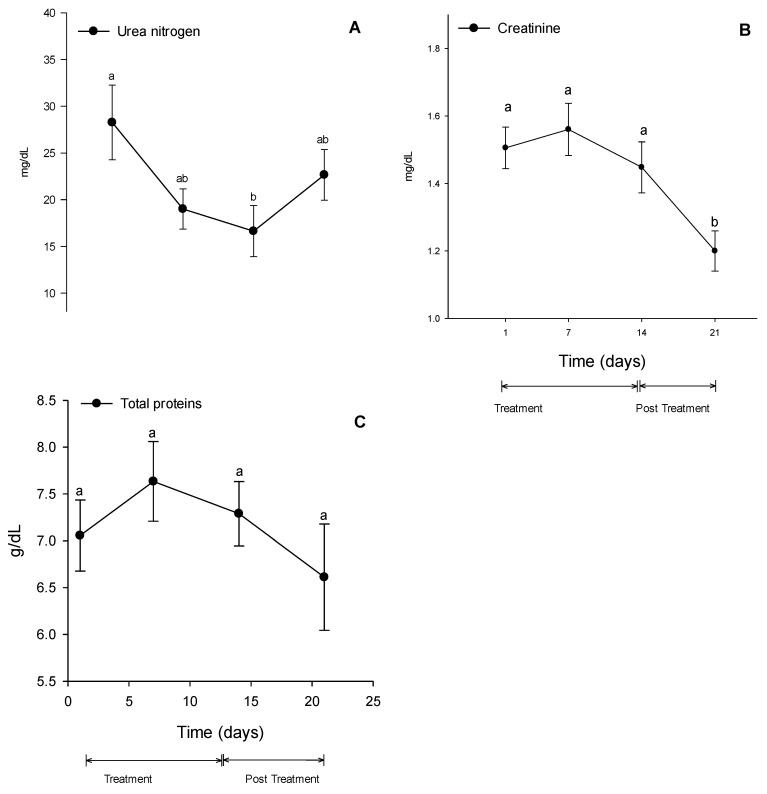
Serum concentrations of urea nitrogen (**A**), creatinine (**B**), and total proteins concentrations (**C**). Different lower-case letters (a or b) indicate significant changes in mean comparisons by the Tukey test. Values are expressed as mean ± standard error.

**Table 1 vetsci-10-00531-t001:** The means and standard deviations and the range of reference for the hematological variables were determined at timepoints D1, D7, D14, and D21.

Variable	Moments of Evaluation (Days)				Reference Range
	D1	D7	D14	D21	
RBC (×10⁶/µL)	7.45 ± 1.14	7.78 ± 1.140	7.78 ± 1.06	7.51 ± 1.09	5.5–9.5
Hb (g/dL)	11.38 ± 1.28 ^a^	12.02 ± 1.30 ^b^	12.61 ± 1.17 ^b^	12.67 ± 1.03 ^ab^	8.0–14
Ht (%)	36.54 ± 5.53	37,36 ± 4.29	37.39 ± 3.94	37.61 ± 4.75	24–44
WBC (10³/µL)	8.39 ± 11.40 ^a^	7.70 ± 1.35 ^ab^	7.28 ± 1.18	8.27 ± 1.30 ^a^	6.0–12
Segs (%)	56.44 ± 5.39	59.11 ± 3.18	58.0 ± 4.72 ^a^	60.56 ± 4.33	35–75
Bands (%)	1.78 ± 0.83	1.89 ± 0.93	2.0 ± 0.71	1.67 ± 0.87	0–2
Lym (%)	36.22 ± 5.61	33.0 ± 3.5	33.67 ± 5.77	32.67 ± 3.04	15–50
Mon (%)	3.11 ± 1.54	2.67 ± 1.15	3.78 ± 1.39	2.89 ± 1.17	2.0–10
Eos (%)	2.44 ± 1.42	3.78 ± 0.97	2.11 ± 1.62	2.22 ± 0.83	2.0–12

Abbreviations: RBC—red blood cells, Hb—hemoglobin, Ht—hematocrit, WBC—white blood cells (leukocytes), Segs—segmented neutrophils, Bands—neutrophils bands, Lym—lymphocytes, Mon—monocytes, Eos—eosinophils. Different lower-case letters (a or b) represent differences in the comparison of means by the Tukey test.

## Data Availability

The raw/processed data required to reproduce these findings are available from the corresponding author upon reasonable request.

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
