# Peer review of "Safety Assessment of an Oral Therapeutic Dose of Firocoxib on Healthy Horses"

_vetsci, 2023, doi:10.3390/vetsci10090531_

Round 1

Reviewer 1 Report

In the article "Evaluation of the Effect of the Oral Therapeutic Dose of Firocoxib on Healthy Horses" the authors describe and discuss the results of the study on the use of Firocoxib in healthy horses. The article is written in an appropriate form. The discussion is satisfactory. The figures are clear and well-described. Congratulations to the Authors of the study.

 I suggest to add:

 2.1

Please provide the number of the permission for use the animals in the experiment

2.3

Please provide the brand and the model of the gastroscope used in the study

2.6

Please provide the brand and the model of aperture used for blood analysis

Reviewer 2 Report

Dear Authors, please include the reference ranges in SI units including the results and discuss the urinanalysis. The paper is attached.

Reviewer 3 Report

Interesting and well done sti=udy.

Please add brands for different m=machines used for the analysis, such as endoscope, and machinery used for blood evaluation.

Please add why increased in fibrinogen and AST could have happened.

In the conclusion add that the drug was given for 21 days.
